# The Conspicuous Link between Ear, Brain and Heart–Could Neurotrophin-Treatment of Age-Related Hearing Loss Help Prevent Alzheimer’s Disease and Associated Amyloid Cardiomyopathy?

**DOI:** 10.3390/biom11060900

**Published:** 2021-06-17

**Authors:** Sergey Shityakov, Kentaro Hayashi, Stefan Störk, Verena Scheper, Thomas Lenarz, Carola Y. Förster

**Affiliations:** 1Department of Anaesthesiology, Intensive Care, Emergency and Pain Medicine, University Hospital Würzburg, D-97080 Würzburg, Germany; shityakoff@hotmail.com; 2Infochemistry Scientific Center, Laboratory of Chemoinformatics, ITMO University, 191002 Saint-Petersburg, Russia; 3Advanced Stroke Center, Shimane University Hospital, 89-1 Enya, Shimane, Izumo 693-8501, Japan; kentaro@med.shimane-u.ac.jp; 4Comprehensive Heart Failure Q9 Center, University of Würzburg, D-97080 Würzburg, Germany; Stoerk_S@ukw.de; 5Department of Otolaryngology, Hannover Medical School and Cluster of Excellence “Hearing4All”, 30625 Hannover, Germany; scheper.verena@mh-hannover.de

**Keywords:** Alzheimer’s disease, amyloid cardiomyopathy, heart failure, age-related hearing loss, neurotrophins, blood–brain barrier, blood–labyrinth barrier, spiral ganglion neuron, BDNF, GDNF

## Abstract

Alzheimer’s disease (AD), the most common cause of dementia in the elderly, is a neurodegenerative disorder associated with neurovascular dysfunction and cognitive decline. While the deposition of amyloid β peptide (Aβ) and the formation of neurofibrillary tangles (NFTs) are the pathological hallmarks of AD-affected brains, the majority of cases exhibits a combination of comorbidities that ultimately lead to multi-organ failure. Of particular interest, it can be demonstrated that Aβ pathology is present in the hearts of patients with AD, while the formation of NFT in the auditory system can be detected much earlier than the onset of symptoms. Progressive hearing impairment may beget social isolation and accelerate cognitive decline and increase the risk of developing dementia. The current review discusses the concept of a brain–ear–heart axis by which Aβ and NFT inhibition could be achieved through targeted supplementation of neurotrophic factors to the cochlea and the brain. Such amyloid inhibition might also indirectly affect amyloid accumulation in the heart, thus reducing the risk of developing AD-associated amyloid cardiomyopathy and cardiovascular disease.

## 1. Introduction: Alzheimer’s—A Systemic Disease?

Alzheimer´s disease (AD) is a neurodegenerative disorder characterized by neurovascular dysfunction [1], cognitive decline [2], accumulation of amyloid β peptide (Aβ) in the brain [3] and the formation of tau-related lesions in neurons called neurofibrillary tangles (NFTs) [4]. AD is the most common cause of dementia in the elderly and is estimated to affect approximately 14 million people in the United States by 2050 [5]. Different theories about the etiology and pathophysiology of AD have been advanced [6]. In the amyloid hypothesis, Aβ plaques are causally involved in cognitive decline [7,8], and elevated levels of Aβ in brain parenchyma may induce or promote neurovascular [9,10] and neuronal dysfunction [11]. This can lead to a cycle of self-propagation [12,13,14], as in a prion disease [15] and ultimately to cerebral β-amyloidosis [16]. The tau hypothesis postulates that abnormalities in the microtubule-associated tau protein initiate the disease by forming NFTs [17]. The neurovascular or “two-hit” hypothesis [18,19,20] claims that two independent pathophysiological events contribute to disease manifestation. First, a cerebrovascular event (hit 1) compromises the blood–brain barrier (BBB), which is critical for brain Aβ homeostasis [18], and then cerebral Aβ accumulation constitutes the second insult (hit 2). Furthermore, many of the currently available drug therapies are based on the cholinergic hypothesis, in which AD is said to be caused by cholinergic effects such as the reduced synthesis of the neurotransmitter acetylcholine [21,22,23]. Very recent reports put forth yet another theory: AD is triggered by dysfunctional β-amyloid precursor protein (APP) metabolism or presenilin (PS) mutations. The subsequent accumulation of APP C-terminal fragments may then lead to overproduction and reduced clearance of Aβ [24]. Finally, there are hypotheses that underscore the importance of environmental risk factors such as air pollution, including smoking, and infections.

While AD is the most common cause of dementia in the elderly, age-related hearing loss (ARHL) is the most prevalent sensory deficit, and growing evidence links it with cognitive decline and an increased risk of dementia. Moreover, it has recently been proposed that AD is a risk factor for heart failure, potentially driven by cardiac amyloidosis. In our review, we set out to present AD as a systemic disease potentially driven by ARHL by highlighting four factors: (i) several diseases and risk factors involving systemic metabolic dysfunction have been linked to an increased risk of developing AD; (ii) no cause-and-effect relationship between these aspects is precisely defined; (iii) common components affecting both the CNS and systemic processes add to AD ontogenesis; and (iv) AD likely represents a multifactorial disease displaying both central and peripheral phenomena.

Specifically, we connect hearing loss to AD development and ultimately to cardiac amyloidosis, while presenting evidence that ARHL is a possible biomarker and modifiable risk factor for cognitive decline, cognitive impairment, and dementia. Moreover, we discuss innovative pursuits to inhibit the Aβ aggregation process and deposition in brain and heart more efficiently through targeted neurotrophic factor supplementation to the cochlea to prevent AD and the development of AD-related amyloid cardiomyopathy.

## 2. The Blood–Brain Barrier

The BBB is formed by the cerebral microvascular endothelium, which tightly controls the homeostasis of the central nervous system (CNS). It consists of endothelial cells (EC), astrocyte end-feet, and pericytes. The basal membrane is responsible for the protection and homeostasis of the brain parenchyma. Chiefly, the tight junctions (TJ) at the zonula occludens in brain capillary ECs create a paracellular barrier that protects the brain from neuroactive substances and other endobiotics and xenobiotics as well as from the external environment. Two different classes of integral membrane proteins constitute the TJ strands in epithelial and endothelial cells: occludin and members of the claudin protein family. In addition, cytoplasmic scaffolding molecules associated with these junctions regulate diverse physiological processes like proliferation, cell polarity and regulated diffusion [25,26,27,28,29], (Figure 1). The BBB is part of the so-called neurovascular unit, which consists of neurons, endotheliocytes, and astrocytes [30] and connects neurons with the bloodstream. The astrocytes are centrally positioned to mediate interactions between neurons and the cerebral vasculature to maintain BBB integrity and neurovascular coupling. In AD, inflammation within the neurovascular unit promotes the apoptosis of astrocytes, shutting down the neuronal food supply and causing BBB impairment and dysregulation. Moreover, the Aβ impact on the vascular component is defined by Aβ deposition not only in the heart but also in the blood vessels, which leads to the propensity for vessel rupture [31]. This condition was previously described as cerebral amyloid angiopathy, a common cause of lobar intracerebral hemorrhage and cognitive impairment [32].

Inflammation is a central mechanism in AD development [34]. Under inflammatory conditions and owing to the specialized structure of the BBB, immune cell entry into the CNS parenchyma involves two consecutive, separately regulated steps: migration of immune cells across the BBB or the blood–cerebrospinal fluid barrier (BCSFB) into the CSF-drained spaces of the CNS, followed by progression across the glia limitans into the CNS parenchyma. So far, research has focused mainly on elucidating the distinct molecular mechanisms required for immune cell migration across the different CNS barriers in multiple sclerosis [35], brain cancer [36] and ischemic brain injury [37], while many other neuroinflammatory diseases and comorbidities like AD and AD-related amyloid cardiomyopathy still await a more profound mechanistic investigation.

The inflammation-associated dysfunction of the BBB in AD has been reported to lead to impaired clearance of neurotoxic Aβ, causing cognitive decline [38,39]. Cerebrovascular dysfunction appears to be one of the key underlying factors of AD pathogenesis, resulting in cerebral hypoperfusion. Likewise, elevated expression of APP [40,41] is thought to result in further Aβ accumulation, predominantly in close proximity to cerebral blood vessels and is detectable in the atherosclerotic intima in 35–60% of patients [42]. Likely contributors to protracted damage to the cerebral vasculature (also for AD) are the reduced activity of the Pgp-and ATP-binding cassette (ABC) efflux transporters involved in Aβ clearance [43,44], altered expression of barrier-constituting TJ proteins [45], cerebral microbleeds, and vasospasms [18,19,20,46,47,48,49,50,51].

Accumulating evidence suggests that a mixed vascular pathology and small-vessel disease contribute to the ontogeny of AD [19,52]. Moreover, reduced brain–blood perfusion [53], silent infarcts [54], and the presence of one or more infarctions [55] all increase the risk of developing AD, which appears to be caused by various, often modifiable, vascular risk factor affecting BBB integrity [26].

### BBB Dysfunction and Vascular Risk Factors for AD

Epidemiological evidence suggests that cerebrovascular disorders and sporadic late-onset AD carry overlapping common risk factors [56]. The combination of increased BBB permeability, inflammatory processes and oxidative stress is an acknowledged triad fueling the development of general dementia [49,57,58]. Generally, diabetes, obesity, hypertension and smoking contribute to inflammation, which fosters the development of AD by promoting BBB dysfunction and increasing its permeability [59]. Therefore, it might be hypothesized that this situation could trigger Aβ permeation across the BBB into the bloodstream. Additionally, reduced levels of low-density lipoprotein receptors at the BBB contribute to elevated Aβ concentrations in plasma [60]. As a result, the subsequent Aβ accumulation and deposition occurs in the extracellular region of cardiomyocytes, leading to heart pathologies such as myocardial infarction, cardiomyopathy, and heart failure. Our hypothesis was confirmed by the preliminary findings of Trancone et al., where the authors observed secondary amyloidosis of the heart of AD patients [61].

Furthermore, hypertension is a key risk factor for developing AD because it induces vascular changes that promote atrophy and increase NFT deposition and Aβ plaque formation [62,63,64,65]. Moreover, mid-life diabetes [66,67,68,69], and obesity [70,71,72] have been shown to increase the risk for both AD and vascular dementia. Amongst the induced vascular changes, neuroimaging, post-mortem analyses and ultrastructure observations could delineate phenomena such as inflammation-induced basement membrane thickening, reduction or degeneration of pericytes that stabilize the capillaries of the BBB and even the accumulation of erythrocytes, resulting in increased BBB permeability and cognitive decline [64].

Diabetes: The diagnosis of type-2-diabetes mellitus (T2M) has been positively correlated with an increased risk of developing AD [67,68,69]. Animal experiments revealed increased and exacerbated cognitive impairment with increased insulin resistance, which is a hallmark of the disease. Increased mitochondrial damage from T2M, thought to amplify the production of cell-toxic ROS, is under investigation as a potential pathophysiological marker [69,73].

Obesity: Elevated plasma levels of cholesterol and free fatty acids are suspected not only of promoting a surge in production and deposition of tau and Aβ proteins, but presumably also of pro-inflammatory cytokines. In addition, these might lower the production or expression of matrix metalloproteinases involved in Aβ clearance resulting in BBB dysfunction. A dysfunctional BBB, however, facilitates the accumulation of harmful neurotoxic endo- and xenobiotics, which cause neuronal damage or death, contributing to cognitive decline [71,72].

Atherosclerosis in the carotid arteries, which supply blood to the brain, increases the risk of dementias, most prominently AD, through the buildup of plaque in the intimal medial thickness region [42,43,58,74,75] (Figure 2).

Morphologically, carotid atherosclerosis is associated with an increased risk of developing dementia [75] because enhanced intima-media thickening and the formation of plaque promote accelerated and pronounced cognitive decline [42,43,58,74]. Figure 2 gives an example of Congo red staining of amyloid deposits in a typical carotid artery endarterectomy specimen.

Stroke: A significant association between stroke and the development of AD has been acknowledged because it enhances BBB permeability, inflammation and oxidative stress [49,57,58].

All the abovementioned risk factors contribute to AD development by inducing vascular dysfunction, but for AD and BBB dysfunction, the chicken or the egg dilemma arises.

## 3. Amyloid Cardiomyopathy

Age-related functional decline in brain and heart function is a typical feature of aging, and a major factor affecting quality of life, psycho-emotional stability, physical strength and self-determination in the elderly [76]. While AD is the fifth-common cause of death in people older than 65, heart failure is responsible for more than one-third of deaths related to cardiovascular disease [61,77]. Heart failure and dementia associated with aging might share common risk factors, mainly related to cardiovascular dysfunction [61]. If so, it would support the assumption that AD is not limited to brain but is a systemic condition requiring appropriate diagnostic approaches and treatments for both brain and heart disease. Although a causal link has not yet been clearly established, multiple reports describe amyloid cardiomyopathy as a prognostically relevant comorbidity for AD. This raises the possibility of a novel subtype of amyloid cardiomyopathy originating from Aβ inclusions in the myocardial tissue of AD patients. Typically, amyloid cardiomyopathy, a progressive infiltrative condition, is caused by either misfolded monoclonal immunoglobulin light chains (ALs) from an abnormal clonal proliferation of plasma cells that exert direct toxic effects on myocytes or by the abnormal folding of transthyretin (TTR), a liver-synthesized protein (previously called prealbumin) that is involved in the transportation of the hormone thyroxine and retinol-binding protein. The result of the latter is the so-called transthyretin amyloidosis (ATTR). The respective pathomechanisms of ATTR and AL amyloidosis that affect the heart are essentially different [78]. Common risk factors for the development of amyloid cardiomyopathy seem to include, besides a genetic predisposition, the biochemical effects of ROS formation, inflammation, abnormal coagulation, or silent strokes [79,80]. Interestingly, it is well-known that AD patients often develop amyloid cardiomyopathy and peripheral and autonomic neuropathies [81,82]. The latter pathology may lead to autonomic cardiac dysfunction/neuropathy that manifests as sinus tachycardia, exercise intolerance, orthostatic hypotension, abnormal blood pressure regulation, dizziness, asymptomatic myocardial ischemia, or infarction [83,84].

In the proposed novel AD-related subtype of amyloid cardiomyopathy, the observed HF symptoms are supposed to be the consequence of amyloid deposition in the myocardial tissue. The prevalence of amyloid cardiomyopathy in elderly patients with otherwise unexplained systolic HF or conduction disorders has been estimated at 11% (95% CI 0–23%) and 2% (95% CI 0–4%), respectively. [85]. In a cross-sectional study, a positive association between an Aβ brain burden measured in vivo and diastolic blood pressure was presented, indicating a possible link between this cardiovascular risk factor and the Aβ burden as measured by Pittsburgh Compound B-positron emission tomography (PiB-PET) [86]. Amyloid deposits in a typical carotid artery endarterectomy specimen associated with AD are shown in Figure 2. Carotid endarterectomy is a surgical technique that removes plaque build-up from inside a carotid artery, i.e., from the bloodstream at the neck. It aims to restore normal blood flow to the brain, thereby preventing a stroke. Proof of a direct connection between heart failure and AD is still outstanding, but there are synergistic pathomechanisms between AD and aging because both conditions induce wall thickening and diastolic dysfunction [79,80,81,82,87]. Patients with AD were shown to exhibit electrocardiographic and echocardiographic abnormalities, including diastolic dysfunction, which reproduces the pattern of cardiac amyloidosis [87]. This strengthens the hypothesis that there may be subclinical cardiac involvement in AD that is likely associated with Aβ amyloid deposition [87]. Gianni et al. first linked cardiac amyloidosis and idiopathic dilated cardiomyopathy (iDCM) by describing a particular pattern of myocardial protein aggregates resembling the tangles and plaques commonly found in AD; however, no overexpression of Aβ was detected [88]. These findings were corroborated when it was shown that these aggregates closely resembled and were biochemically equivalent to, those found in the brains of AD patients [89]. Extending these findings even further, genetic variants in the gene associated with early-onset AD (presenilin-1) were identified in familial and sporadic cases of iDCM [88,90]. Of note, amyloid in the heart was not limited to AD deposition; that is, it was not secondary to the pathology of another organ; rather, it appeared to be caused by a common underlying mechanism, e.g., systemic light chain AL-amyloidosis [91] or transthyretin amyloidosis [61,92].

A pioneering study that monitored the presence and characteristics of structural and functional cardiac abnormalities in AD patients, was presented by Troncone et al. They investigated the mechanism by which Aβ accumulation in the heart interfered with heart function [61]. Using imaging and proteomics, they demonstrated that Aβ aggregates can propagate outside the brain, reach the heart and induce AD-related cardiac amyloidosis [61]. Thus, the toxic effects of Aβ pre-amyloid oligomers (PAOs) on cardiomyocytes were demonstrated [88], thus causally linking cardiac amyloidosis with AD-related heart failure. This supports the notion that, as with the brains of AD patients, Aβ40 and Aβ42 amyloid plaques and NFTs formed by abnormally hyperphosphorylated microtubules associated with the tau protein constitute a key feature of amyloid cardiomyopathy associated with AD. In this study, Aβ amyloid aggregates were identified both within cardiomyocytes and in interstitial samples from the dysfunctional myocardia of 4 AD patients [61]. Compared to controls, patients with AD exhibited thicker ventricular walls yet smaller end-diastolic and end-systolic diameters, which were compatible with compromised stroke volume and diastolic function. These observations supported the notion that Aβ amyloid accumulation in the heart contributes to myocardial dysfunction in AD-linked cardiac amyloidosis. In follow-up experiments, the impaired function of cardiomyocytes was demonstrated experimentally and confirmed in AD patients. A peculiar involvement of the heart in AD became apparent, characterized by a generally mild increase in interventricular septum thickness, a higher degree of diastolic dysfunction, and ECG abnormalities, including reduced QRS voltage or voltage/mass ratio [87]. Consistently, cardiomyocytes of affected patients displayed relaxation deficits, which likely originated from extracellular Aβ infiltrates. In summary, Aβ aggregates were identified both within cardiomyocytes and in the myocardial interstitium of AD patients, proving that Aβ oligomers derived from brain-originating protein misfolding can reach, and may cause damage to, the heart [3,87,92]. It has, however, not been unraveled yet how Aβ may cross the BBB to induce systemic forms of amyloidosis like cardiac amyloidosis.

Although this data suggest that protein misfolding may affect both the brain and the heart in AD patients, the clinical relevance of this finding still needs to be demonstrated as the sample size of respective clinical studies was very small. Hence, clinicians should attentively monitor cardiac abnormalities when evaluating patients with AD to clarify the full spectrum of myocardial involvement and its comprehensive clinical relevance. Furthermore, the question whether these conditions are part of a common multi-organ syndrome or are causally linked to disparate conditions remains to be answered. Current evidence suggests that AD and HF may be both dependent and independent of each other, but the coexistence of AD and HF points to a new sub-class of amyloid cardiomyopathy with a potentially large impact on quality of life expectancy.

## 4. Current and Novel Treatment Modalities

Current pharmaceutical approaches targeting neuropathologic processes such as AD offer limited, predominantly symptom-modifying, effects [93,94] but have failed so far to prevent further neurodegeneration. The available therapeutic targets are limited to two candidates that have been shown to improve cognitive function: acetylcholinesterase (AChE) inhibitors and N-methyl-D-aspartate (NMDA) receptor antagonists.

### Novel Inhibitors of Amyloid Fibril Formation in AD

The pathological process in AD is based on the formation of amyloid fibrils (Aβ) and amyloid oligomers, which have to be explored to inhibit the Aβ aggregation process more efficiently [95,96]. Therefore, by developing and applying the appropriate techniques for measuring Aβ affinity to drug-like and supramolecular structures in the solution, it is possible to discover novel therapeutics for AD. Previously, the amyloid β-sheet breaker was determined to be N,N’-bis(3-hydroxyphenyl)pyridazine-3,6-diamine (RS-0406), which is capable of significantly inhibiting 25 mM Aβ fibrillogenesis (Figure 3a,b), thus suggesting that RS-0406 or one of its derivatives could become a therapeutic agent for AD patients [97].

Moreover, atomic force microscopy, together with molecular dynamics simulations, has demonstrated that some pseudopeptides might bind to amyloid fragments with varying degrees of affinity to prevent Aβ–Aβ aggregation [98]. Similarly, the other β-sheet breaker peptides were designed to complement the enthalpic interactions with the aggregating protein to inhibit the pathological process in a concentration-dependent manner [99]. Additionally, the neuroprotective effect of single-wall carbon nanotubes with built-in peroxidase-like activity against Aβ-induced neurotoxicity was established to diminish the formation of Aβ fibrils [100].

On the other hand, the cyclodextrin-based formulations of different drug-like molecules were also developed in many previous studies [101,102]. Some modified cyclodextrins, such as hydroxypropyl-beta-CD (HPβCD), have already been implemented as inhibitors in cell and mouse models of AD [103,104]. In particular, the subcutaneous administration of HPβCD to Tg19959 mice significantly improved memory deficits and reduced amyloid deposition, microgliosis, and τ-immunoreactive dystrophic neuritis. These effects occurred, at least in part, by reducing the amyloidogenic processing of APP and enhancing ABCA1-mediated Aβ clearance. The present data suggest that HPβCD may have therapeutic potential for treating AD, but more research is needed to develop efficient and cytotoxicity-free excipients with improved pharmacokinetics and pharmacodynamics (Figure 4).

In a transgenic mouse model of AD, the monoclonal antibody aducanumab was shown to target Aβ plaques selectively [105]. This compound entered the brain, bound parenchymal Aβ, followed by the reduction of its soluble and insoluble variants in a dose-dependent manner [105]. In summary, compounds such as peptide inhibitors, carbon allotropes, biologicals (antibodies), small molecules, and supramolecular complexes that target specific Aβ subregions represent the first generation of amyloid-based therapeutics with the potential to demonstrate disease-modifying activity. Gaining additional insights into amyloid biology and AD will likely guide the development of the next generation of inhibitors. A very promising recent radiation pilot study could provide evidence that treatment with a low dose of radiation in severe AD candidates could significantly improve quality of life. It is speculated that it would cause minor damage to molecules suitable for stimulating a cellular protective response involving antioxidant production and cellular damage repair mechanisms [106].

Alternatively, switching to a preventive strategy through the reduction of risk factors may be more appropriate than pharmacotherapy after the clinical expression of neuropathologic changes [107,108] and could lead to significant reductions in medical costs [109].

## 5. Age-Related Hearing Loss—A Risk Factor for Dementia and AD?

Sensory changes, particularly impaired hearing and vision, and motor changes that contribute considerably to cognitive symptoms, have been classified by the US National Institute on Ageing, as so-called modifiable risk factors for AD [110]. Modifiable risk factors are treatable medical conditions, and lifestyle choices have their own biological mechanisms and contribute to AD etiology and pathophysiology [111]. A plethora of conditions (e.g., cardiovascular disease, type 2 diabetes mellitus, traumatic brain injury, epilepsy, depression, lack of physical activity, sleep disturbances, dietary factors, smoking, alcohol abuse [111], and ARHL) have been identified as affecting AD severity. Amongst those, we focused on the role of ARHL in the development of AD and AD-related amyloid cardiomyopathy.

ARHL (sometimes referred to as presbycusis) is the third most common chronic health condition in the elderly and correlates positively with the risk of cognitive impairment and dementia [5]. More than 30% of those older than 65 are estimated to be suffering from disabling ARHL, and it its effects progress as they age: roughly 65% of people over 70 years and even 80% of people over 85 years present with ARHL [5].

Progressive hearing impairment leads to social isolation, and to comorbidities like frailty, falls, and late-onset depression, ARHL has been linked with cognitive decline and an increased risk of dementia [13]. (“Frailty” describes yet another age-related multidimensional clinical condition characterized by nonspecific vulnerability, reduced multisystem physiological resilience, and decreased resistance to stress leading amongst others to an increased risk of falls, institutionalization, hospitalization, disability, and death [14].) Based on this finding, the US National Institute on Aging has rated ARHL as a possible biomarker and modifiable risk factor for AD.

ARHL can be described as a progressive, bilateral, symmetrical yet multifactorial disorder that affects hearing sensitivity, mostly in the high-frequency spectrum [112]. The major cause is a lifetime of insults to the auditory system [113], involving not only the inner ear via strial damage, loss of hair cells and spiral ganglion neuron (SGN) degeneration (leading to a decreased hearing threshold), but also the central neural auditory pathways, resulting in impaired discrimination, especially in noisy environments [114]. Moreover, this classifies ARHL as a so-called marker for frailty in aged individuals.

### Neurovascular Dysfunction at the Root of ARHL and AD?

AD and ARHL appear to share a common underlying pathogenesis linked to age-related neurovascular dysfunction: dysfunction of the BBB [64] and blood–labyrinth barrier (BLB) [115], which supply the brain and the inner ear, respectively. In the cochlear lateral wall––the barrier between the vasculature and the inner ear fluids critical for auditory and vestibular function––the BLB maintains ion homeostasis, transport of nutrients and systemic hormones to the inner ear [115]. Its neurovascular qualities and similarities to the BBB have been recently described [116]. Accordingly, it appears that age-induced alterations in BLB integrity might promote ARHL [117], so the preservation of both BLB and BBB integrity might help prevent AD and ARHL by maintaining respective tissue homeostasis.

Following the central AD dogma that tauopathy precedes Aβ deposition [114], the authors stress that hyperphosphorylated tau proteins are more frequently reported in the brain’s central auditory pathway long before Aβ deposition starts. This offers striking evidence for early NFT formation [112] in contrast to explaining the rise of both pathologies by considering ARHL as one factor inducing AD tauopathy and APP deposition as the link between ARHL and AD. Correlations between tauopathy and AD–ARHL need to be elucidated to identify the origin of the cascade and to establish appropriate therapies.

We concluded from this that the identification of common mechanisms underlying the epidemiological association between ARHL and AD carries significant implications. We speculate that this association might be related to age-dependent decline in the structure and function of the BLB and BBB, which places neurovascular decline at the root of both ARHL and AD, respectively.

This hypothesis is further strengthened by the fact that many neurovascular diseases are linked to frailty (e.g., stroke, congestive heart failure, diabetes-associated complications) seem to include hearing impairment as a comorbidity [113].

## 6. Neurotrophic Factors at the Interface of AD Pathophysiology and Neural ARHL

Sensorineural hearing loss (SNHL) is generally caused by the loss of hair cells, the sound transducing sensory cells of the cochlea, or by loss of peripheral and central parts of spiral ganglion neurons (SGNs) [118], which connect the hair cells to the nucleus cochlearis in the brainstem. Age-related loss of peripheral and central SGN synapses (neural ARHL) is consistently observed in humans and animals and is one of the main contributing factors to the development of ARHL [119,120]. To prevent it, the survival of SGNs during aging depends on both genetic and epigenetic influences, and this has been demonstrated at the systemic, tissue, cellular, and molecular levels [114].

Interestingly, studies on laboratory animals have highlighted that survival of SGNs is dependent on a number of genes encoding for special supplying factors: neurotrophic factors such as brain-derived neurotrophic factor (BDNF) and neurotrophin 3 (NT-3) [121] or glia-cell line derived neurotrophic factor (GDNF) [122].

Neurotrophic factors are a group of secreted proteins in neural and non-neural tissues that have multiple functions such as mediating development, homeostasis and survival of the peripheral and CNS [123].

Of noteworthy significance, AD shows an imbalance in specific paralleling neurotrophic factors and receptors, and the same is indirectly seen in ARHL, where BDNF proteins are drastically reduced in peripheral and central auditory projections [114] and BDNF and tropomyosin receptor kinase B (TrkB)-agonists improve SGN survival [124]. For a review see Leake et al. [125]. The identified neurotrophic factors have been shown to be critical for neuronal degeneration because in both illness they prevented cell death and enhanced the growth and function of neurons affected by the degenerative processes [126,127]. In this context, neurotrophin nerve growth factor (NGF), GDNF and BDNF stand out [128].

Given their importance, we briefly discuss the major findings about the NGF, BDNF and GDNF support necessary for maturation, proliferation and survival of affected neurons in AD and neural ARHL. In addition, we examine their putative targeting in frame of new therapeutic approaches for the management of these diseases.

### 6.1. AD and Neurotrophins

NGF is integral for the development of the cholinergic system, including neuronal survival and differentiation. It is synthesized in the cortex and hippocampus and known to be retrogradely transported to cholinergic neurons in the basal forebrain. When taken into account that these neurons constitute the main cholinergic innervation to the hippocampus and neocortex, they play an essential role in cognition and attention processes; hence, the cholinergic theory of AD had been developed on alterations to this system (compare above). Furthermore, it was reported that cortical GABAergic neurons are the primary source of NGF synthesis, which provides support for basal forebrain cholinergic projections in adulthood [123]. In AD subjects, postmortem tissue analysis showed that NGF levels had decreased in the nucleus basalis of Meynert, a neuronal group that projects large cholinergic innervation to widespread cortical areas and is well known for undergoing degeneration in this disorder [129].

BDNF is largely expressed in the CNS and influences several aspects of the neuronal function. BDNF mediates hippocampal plasticity in adulthood and the survival and integration of hippocampal new-born neurons, assists the early and late long-term potentiation (LTP) phases, and works as cellular substrate for learning and memory. A substantial number of studies support the idea that BDNF is crucial for discovering the etiology of AD (For a review see [130]). According to preclinical reports, AD transgenic mouse models showed decreased cortical BDNF expression as well as BDNF-mediated TrkB impairment of retrograde neuronal signal transport, whereas studies of serum BDNF levels in subjects with either severe AD or mild cognitive impairment reported conflicting results. Alterations in the anterograde and retrograde transport of BDNF-containing vesicles by extracellular products from APP also were recently demonstrated. Additionally, Aβ at a sublethal concentration downregulates BDNF signaling in cultured cortical neurons, while BDNF was shown to be neuroprotective against Aβ-induced apoptosis in neuroblastoma cells [131]. In contrast, both protein and mRNA levels of BDNF were elevated in cells submitted to Aβ-(25–35) treatment.

GDNF supports several neuronal populations in the central nervous system, including mid-brain dopamine neurons and motoneurons. In addition, GDNF promotes the survival, and regulates the differentiation, of many peripheral neurons such as sympathetic, parasympathetic, sensory and enteric neurons [132]. Depletion of GDNF seems to be linked with AD pathology and symptoms [133], whereas in patients with early-stage AD, increased GDNF levels in the cerebrospinal fluid and decreased serum concentration levels suggest that the impaired brain is adaptive [134]. Increased GDNF levels in the plasma of AD patients were also reported [135]. Moreover, in the postmortem middle temporal gyrus of AD patients, mature GDNF peptides were downregulated. In similar fashion, a study found that serum GDNF levels significantly were reduced. But it has to be noted that others did not observe significant differences in the GDNF levels in the plasma of AD patients and control groups [136]. Nevertheless, a growing body of evidence suggests that GDNF levels are probably involved in the pathogenesis and progression of AD, suggesting that its blood levels could potentially be used as an AD biomarker [137].

### 6.2. Neural ARHL and Neurotrophins

NGF, BDNF and GDNF, amongst other factors are relevant for the lifelong development and function of the auditory system. These three factors are known for their protective effects on inner ear structures in different pathological settings in vitro and in vivo.

During the development of the embryo, both NGF and its receptors are widely expressed in the inner ears [138]; levels decrease after birth, and remain low in adulthood [139]. NGF is involved in protecting degenerating neurons and guiding the growth of nerve fibers, including auditory nerve fibers, toward their target tissues. Its serum levels are reduced in patients with SNHL, and NGF therapy results in SGN preservation in mouse models of ARHL [140].

In the adult inner ear, hair cells and supporting cells as well as neurons and satellite cells are involved in BDNF production, release and homeostasis [141]. BDNF imbalance is observed in ARHL as shown in rat and gerbil animal models where a significant reduction of BDNF transcripts in high-frequency processing cochlear neurons was observed during aging, but this did not coincide with a major reduction in BDNF protein. In contrast, it was drastically reduced in peripheral and central projections [124]. On the other hand, an age-related increase in BDNF was observed in the organ of Corti, but not in the cochlear ganglia [142]. An exogenous BDNF supply was recently proven to reduce inner hair cell–SGN synaptopathy in vitro [143] and has been extensively and successfully studied in vitro and in vivo for its protective effects on SGNs [144,145].

The GDNF and its receptors are expressed in the sensorineural epithelium and the lateral wall and SGN of the cochlea. Throughout the process of aging, the expression of this neurotrophic factor and the GFRα-1 protein, its high-affinity receptor, is affected in different target tissues. Yurek et al. measured the GDNF in the striatum and ventral midbrain of young and aged rats following a unilateral lesion of the nigrostriatal pathway. The endogenous GDNF was affected with one noticeable difference: the lesion in the young brain displayed a significant compensatory increase in the denervated striatum, while no compensatory increase was observed in the aged brain [146]. In the aged skin, the GDNF and its receptors were reduced as well. The in vitro supply on dissociated auditory neurons and local inner-ear therapy in animal models proved to have a strong protective effect on SGNs [147].

The proteins NGF, BDNF and GDNF and their receptors are expressed in the CNS and the inner ear and are affected by age as described above. Since AD and ARHL are both affected by neurotrophic factor therapy, the already well-known hypothesis that ARHL affects AD may be true. We now present a second hypothesis: the age-related imbalance of neurotrophic factors may be a link between AD and ARHL––not in the sense of AD being affected by ARHL, but as a fundamental pathology such as neurovascular dysfunction.

Which pathology subsequently affects the other may be irrelevant in view of the therapeutic effect of the neurotrophic factor therapy. Since the supply of neurotrophic factors preserves SGNs from degeneration, they reduce the anatomical correlate of neuronal ARHL, so there could emerge a potent therapeutic option in preclinical models to treat AD and neural ARHL at the same time. These therapies show promise for protecting or stimulating regeneration in hair cells and SGNs, thus regenerating or even preventing deafness of the mammalian inner ear [130] and having a positive parallel effect on AD. However, their effectiveness in clinical studies remains to be demonstrated since safety and efficacy are not yet proven [125,131,132]. Additional research and randomized clinical trials are warranted to examine the treatment implications for cognition and ARHL and to explore possible causal mechanisms underlying their relationship.

## 7. Conclusions and Open Questions

The aforementioned sections highlight the central role that investigations into neurovascular dysfunction in AD and ARHL have taken and highlight a number of interrelated mechanisms that may contribute to AD, AD-related amyloid cardiomyopathy and ARHL pathogenesis in a likely cascade at the brain–ear–heart axis (Figure 5).

As the literature demonstrated, neurovascular dysfunction accelerates core ARHL and AD pathologies; therefore, investigations into preventive and therapeutic approaches that target neurovascular dysfunction should focus on the root of both illnesses. Extensive investigations of AD patient populations as well as multi-component, pre-clinical model systems will be needed to further evaluate and characterize cardiac amyloidosis arising from the systemic effects of AD pathology to prevent AD-related heart failure. Given recent developments, ARHL might be a modifiable risk factor for AD development which could have a global impact. The underlying neural presbycusis might be preventable by targeting neurotrophin transport to the cochlea. Suggestions to preclude ARHL, AD and AD-related amyloid cardiomyopathy through timely treatment of SGN loss by neurotrophin supplementation has become an important topic in potential treatments of AD and its comorbidities.

## Figures and Tables

**Figure 1 biomolecules-11-00900-f001:**
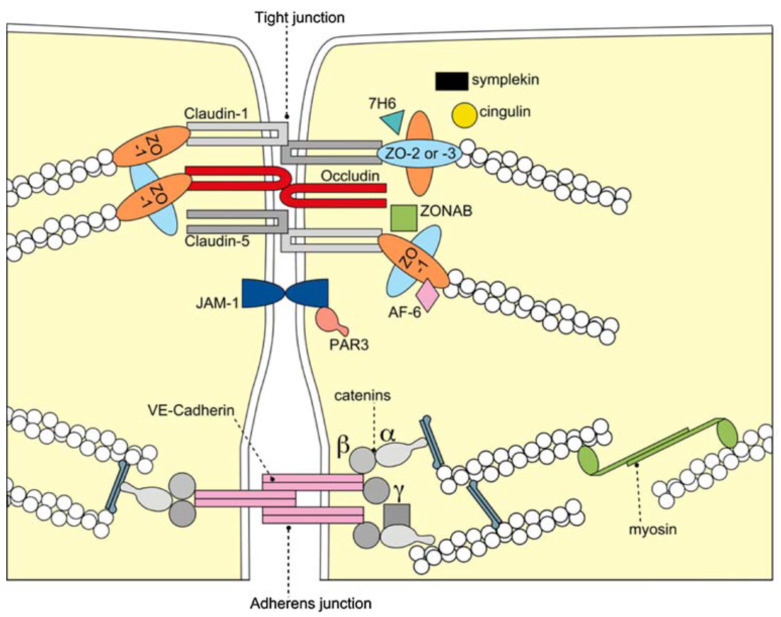
Molecular composition of TJs at the BBB. The transmembrane proteins occludin, the claudin family, and junctional adhesion molecule-1 (JAM-1) constitute the barrier formed by the sealing of the paracellular space by TJs. They appear to interact in a homophilic manner, and occludin seems to co-polymerase into claudin-based TJ strands. BBB = blood brain barrier; TJ = tight junction; VE-cadherin = vascular endothelial cadherin, ZO-1,-2,-3 = zonola occuludens protein-1,-2,-3. Adapted from [33] with permission.

**Figure 2 biomolecules-11-00900-f002:**
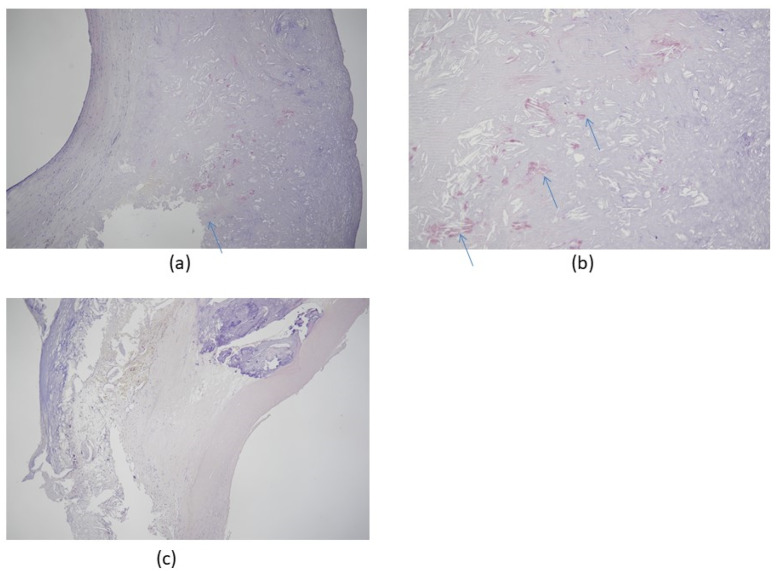
Congo red stain of carotid artery atherosclerosis dissected by carotid endarterectomy. The specimen was fixed in 4% formalin after surgical removal and stained with Congo red for amyloidosis, which colored the Aβ protein plaque deposits red/salmon pink. (**a**) Congo red stain positive for amyloidosis, 50×; (**b**) Congo red stain positive for amyloidosis, 100× (hypermagnification of a); (**c**) negative control, 50×. All patients were scheduled for elective therapeutic endarterectomy and gave informed written consent to the procedure. Arrows indicate amyloid deposits in tissue. Images provided by Hayashi, K., Shimane University Hospital, advanced stroke center.

**Figure 3 biomolecules-11-00900-f003:**
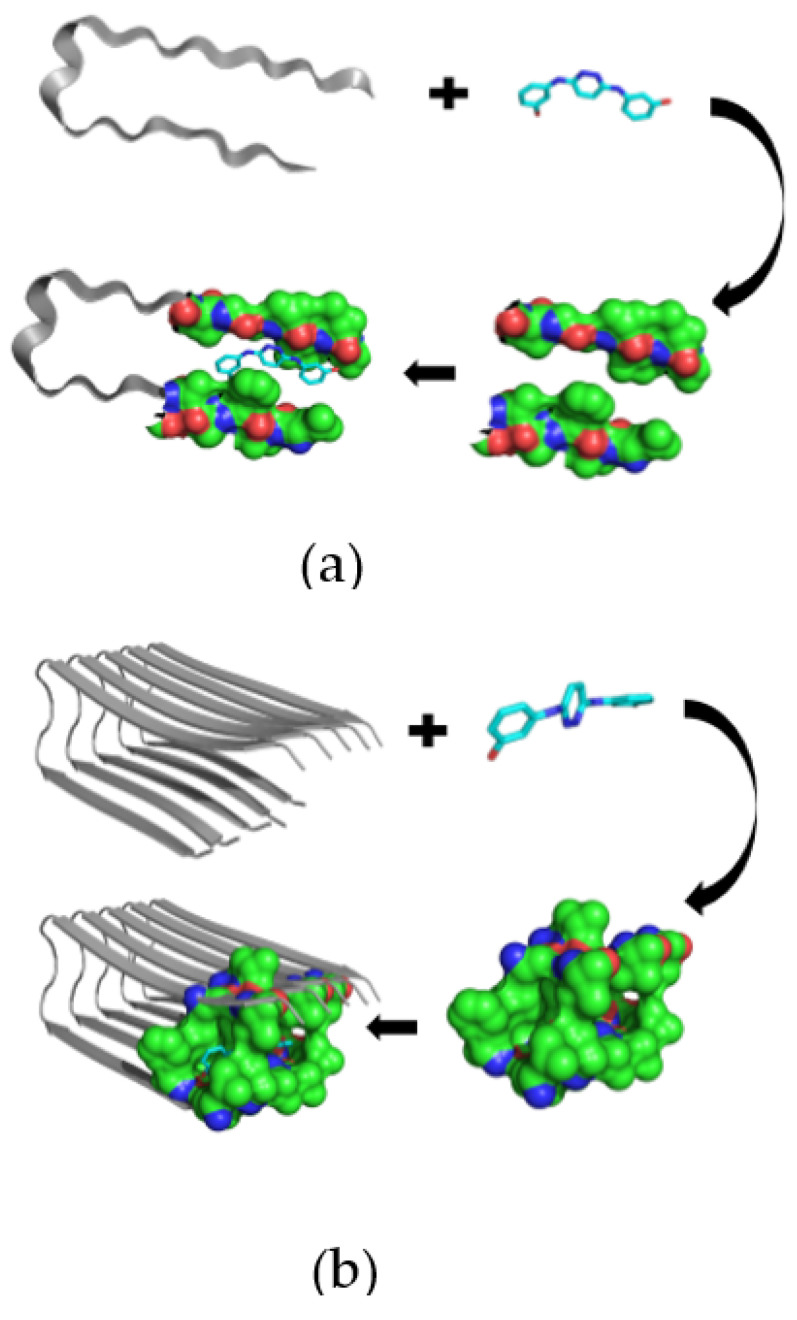
The predicted interaction of the RS-0406 molecule with Aβ as a monomer (**a**) or oligomer (**b**). The binding site is shown as a molecular surface predicted by the CASTp algorithm (Shityakov et al., unpublished).

**Figure 4 biomolecules-11-00900-f004:**
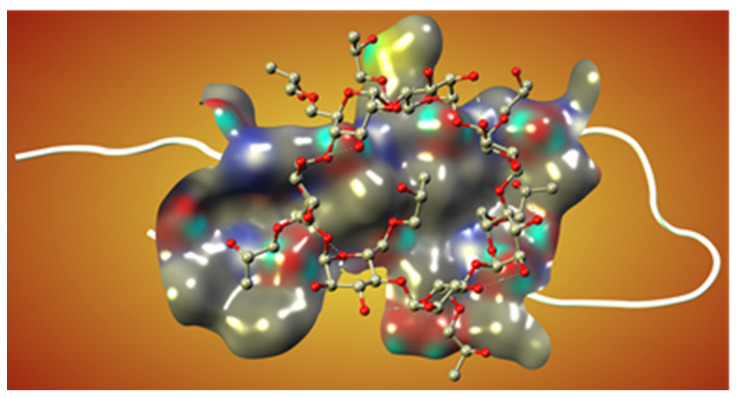
The predicted interaction of the HPβCD molecule with Aβ as a monomer. The monomer is depicted in white, and the cyclodextrin is shown as a ball-and-stick model. The molecular surface represents the interaction between protein and ligand structures (Shityakov et al., unpublished).

**Figure 5 biomolecules-11-00900-f005:**
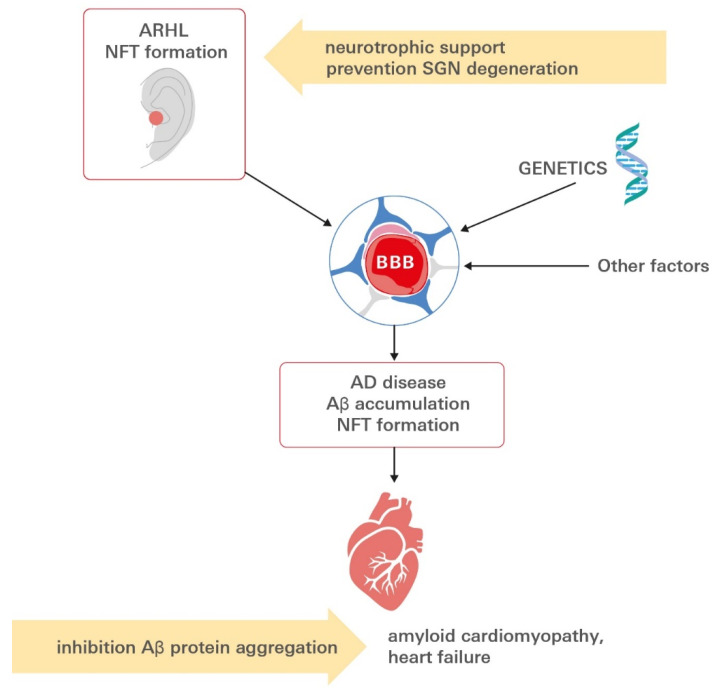
Linking ARHL to the development of AD-related amyloid cardiomyopathy. Aβ aggregates extend outside the CNS to the myocardium in AD. Patients diagnosed with AD present with clinical signs of functional defect, while tauopathy is present long before the onset of symptoms in the auditory tract. Besides Aβ-transport alterations, genetic and other factors might contribute to altered BBB integrity. Aβ deposits can be detected in the hearts of affected subjects, impairing cardiac function, ultimately leading to heart failure. In the context of increasing life expectancy, these findings raise an alarming threat for public health because they suggest that AD, ARHL and AD-related HF may coexist either through a common etiology or as phenotypes of a multiorgan syndrome. Future potential preventive or therapeutic intervention points to the inhibition of Aβ protein aggregation or neurotrophin supplementation to nurture and preserve spiral ganglion neurons. Aβ: amyloid β peptide; AD: Alzheimer’s disease; ARHL: age-related hearing loss; BBB: blood–brain barrier; CNS: central nervous system; SGN: spiral ganglion neuron, HF: heart failure.

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
