# Peer review of "The Conspicuous Link between Ear, Brain and Heart–Could Neurotrophin-Treatment of Age-Related Hearing Loss Help Prevent Alzheimer’s Disease and Associated Amyloid Cardiomyopathy?"

_biomolecules, 2021, doi:10.3390/biom11060900_

Round 1

Reviewer 1 Report

Shityakov et al present a review that draws links between AD and ARHL, and postulates potential disease altering treatment strategies. The authors combine several theories and observations to produce a holistic theory for the pathophysiology and progression of AD. And while the current knowledge and data poses challenges to combining several separate amyloidosis pathways to make a combined theory, the authors do an admirable job. A few minor issues remain.

  • L116-117: This is a strongly formulated sentence that generalizes the origin of AD. All theories at this point are unproven and such generalization and or conclusions are not appropriate.
  • Ending in L217: How do the authors reach this conclusion when ref. 78 specifically notes: “Symptoms and signs of heart failure were absent in all patients with AD.”
  • L218-220: Ref 79 specifically states: “We did not detect differences in the level of Aβ in iDCM compared to controls at immunohistochemistry”. However, “Importantly, similar pathology was found in the core of myocardium removed for LVAD implantation in a patient with end-stage iDCM”. Specifying the subpopulation will clear potential confusion.
  • Paragraph starting at L222: It is important to note the disease stage at which AD patients experienced cardiac abnormalities.
  • In part 4, the authors should mention the radiation pilot study which has shown promising results. DOI: 10.3233/JAD-200620
  • In part 5, authors need to mention that strial damage/degeneration and loss of hair cells are (also) significant factors in ARHL. This is mentioned in part 6, but need to be mentioned in part 5 first; authors can move the explanation in part 6 to part 5 or have it in both locations.
  • Part 6 should be reorganized so that the connection between the different factors is clearer, instead of jumping from one to the other. E.g. P11 gives a somewhat detailed comparison of the different factor in AD and ARHL. However, later, P12 some of the factors and their effects on the auditory system is described. The more logical progression would be to describe the factors for each individual malady and then describe their overlaps. Additionally, since this is the most important section that brings together the different hypothesis, authors should give a more detailed description. They may get inspiration from DOI: 10.3389/fncel.2019.00363 and 10.14336/AD.2015.0825 or similar publications.

Minor issues

  • Grammar and spellcheck is needed. Also check Greek letters.
  • L200, specify that HF is heart failure.
  • In general, abbreviations should be described when used first. All following uses should not elaborate the meaning of the abbreviation.

Author Response

The authors are grateful to Reviewer 1 for providing important comments and critiques which considerably improve the quality of the manuscript.

Alterations were considered as follows:

  • L116-117: This is a strongly formulated sentence that generalizes the origin of AD. All theories at this point are unproven and such generalization and or conclusions are not appropriate.RESPONSE:Thank you. This should of course be avoided. We rephrased the sentence in a more moderate way We also rephrased another generalizing sentence (L130/131): 
  • “There is accumulating evidence that a mixed vascular pathology and small-vessel disease contributes to the ontogeny of AD”.
  • “There is epidemiological evidence that cerebrovascular disorders and sporadic late-onset AD might partly show overlapping common risk factors”.
  •  
  •  
  • Ending in L217: How do the authors reach this conclusion when ref. 78 specifically notes: “Symptoms and signs of heart failure were absent in all patients with AD.”RESPONSE:„Proof of a direct connect between heart failure as such and AD is still outstanding, but there are synergistic pathomechanisms between AD and aging as both conditions induce wall thickening and diastolic dysfunction. Patients with AD were shown to exhibit electrocardiographic and echocardiographic abnormalities, including diastolic dysfunction, reproducing the pattern of cardiac amyloidosis [78]. This strengthens the hypothesis that, in AD, there may be subclinical cardiac involvement likely associated with Aβ amyloid deposition [78].“ 
  •  
  • Thank you for your careful reading. You are correct that all patients reported with AD in ref#78 did not show symptoms of HF. We rephrased this section (starting at L209), which now reads:
  •  
  • L218-220: Ref 79 specifically states: “We did not detect differences in the level of Aβ in iDCM compared to controls at immunohistochemistry”. However, “Importantly, similar pathology was found in the core of myocardium removed for LVAD implantation in a patient with end-stage iDCM”. Specifying the subpopulation will clear potential confusion.RESPONSE„Gianni and coauthors were first linking cardiac amyloidosis and idiopathic dilated cardiomyopathy (iDCM) by describing a particular pattern of myocardiac protein aggregates resembling the tangles and plaques commonly found in AD, although no overexpression of Aβ was detected [79]. These findings were corroborated when it was shown that theses aggregates closely resembled and were biochemically equivalent with those found in the brains of AD patients [80].“ 
  •  
  • Again, thank you for your careful reading. We rephrased the para (starting at L216), which now reads as follows:
  •  
  • Paragraph starting at L222: It is important to note the disease stage at which AD patients experienced cardiac abnormalities.RESPONSE„Of note, patients with AD compared to controls exhibited thicker ventricular walls yet smaller end-diastolic and end-systolic diameters, compatible with compromised stroke volume and diastolic function.“ 
  •  
  • Thank you for this important remark. The original manuscript by Troncone et al, however, does not specify this. There it is stated: „We studied a cohort of AD patients with the clinical diagnosis or diagnostic workup of AD, in the absence of other underlying conditions affecting myocardial function (including history of coronary artery disease [CAD], previous myocardial infarction, hypertension, primary or secondary amyloidosis, dilated/hypertrophic cardiomyopathy, endocarditis, chemotherapy, or radiotherapy). Severity or stage of AD is not specifically mentioned or characterized. However, we added one explanatory sentence characterizing the cardiac morphology of AD patients compared to controls. The respective sentence was introduced in L240 and reads as follows:
  •  
  • In part 4, the authors should mention the radiation pilot study which has shown promising results. DOI: 10.3233/JAD-200620RESPONSE: 
  •  
  • It is very important to mention this study, thank you. We included the following comment: “A recent very promising radiation pilot study could give evidence that treatment with a low dose of radiation in candidates affected with severe AD could significantly improve quality of life. It is speculated that the low dose of radiation would cause minor damage to molecules suitable to stimulate a cellular protective response involving antioxidant production and cellular damage repair mechanisms“.
  •  
  • In part 5, authors need to mention that strial damage/degeneration and loss of hair cells are (also) significant factors in ARHL. This is mentioned in part 6, but need to be mentioned in part 5 first; authors can move the explanation in part 6 to part 5 or have it in both locations.RESPONSE:Thank you for this comment. We agree and decided to mention this very important fact in both sections. Therefore we modified the relevant sentence as follows: 
  • “The major cause for ARHL is the lifetime exposure to insults to the auditory system, involving not only the inner ear via strial damage, loss of hair cells and SGN degeneration (leading to a decreased hearing threshold), but also the central neural auditory pathways (resulting in impaired discrimination, especially in noisy environments)”
  •  
  •  
  • Part 6 should be reorganized so that the connection between the different factors is clearer, instead of jumping from one to the other. E.g. P11 gives a somewhat detailed comparison of the different factor in AD and ARHL. However, later, P12 some of the factors and their effects on the auditory system is described. The more logical progression would be to describe the factors for each individual malady and then describe their overlaps. Additionally, since this is the most important section that brings together the different hypothesis, authors should give a more detailed description. They may get inspiration from DOI: 10.3389/fncel.2019.00363 and 10.14336/AD.2015.0825 or similar publications.
  •  
  • RESPONSE:

Thank you, this is a very important issue! Part 6 has been reorganised, missing references added.

Minor issues

  • Grammar and spellcheck is needed. Also check Greek letters.
  •  
  • RESPONSE:

Was done by Mr. Todd Axel Johnsen, compare acknowledgements.

  • L200, specify that HF is heart failure.
  •  
  • - done

  • In general, abbreviations should be described when used first. All following uses should not elaborate the meaning of the abbreviation.We apologize for being inconsistent and modified the manuscript accordingly. We decided to keep the highlighted „ age-related hearing loss” when first mentioned in section 5 because we believe that the reader will appreciate the repeated introduction in this section.
  •  

Reviewer 2 Report

In this review article (The conspicuous link between ear, brain and heart – could neurotrophin-treatment of age-related hearing loss help prevent Alzheimer’s disease and associated amyloid cardiomyopathy?),  authors tried to brief the interrelationship or link auditory, cardio and neurovascular system during age-related AD condition. The information collected and provided are satisfactory; however, it needs some changes/exploration.
1. The term neurovascular should be defined appropriately: how the neuron can make a blood connection. Brief about the role of the glial cells in neurovascular dysfunction. It may provide a clear picture of how Inflammation (neuroinflammation) participates in amyloid cardiomyopathy, leading to myocardial infarction followed by heart failure during Age-related AD, as Inflammation (neuroinflammation) plays a role significant role in all types of cardiomyopathies, AD-allied pathologies, and BBB permeability.

  1. A few more lines should be added about the cardiac status of the AD patients/animal models and amyloid cardiomyopathy related to peripheral and autonomic neuropathy.
  2. Although the risk factors ofhypertension, obesity etc.mentioned here are the prominent risk factors for cardiovascular diseases, mechanistic relations to the  BBB permeability and AD should be provided, like how the BBB permeability affects the Aβ accumulation/deposition, which could be based on the Aβ transportation/accumulation in the extracellular region of cardiomyocytes, leading to myocardial infarction/hypertrophy and its associated abnormalities.
  3. One of the exciting para/lines (line 378-382) that links the ARHL and AD: whether Tauopathy is the root link between ARHL and AD?
  4. Line 401, the neurotrophins: indicateall neurotrophins or specific.
  5. More evidences arerequired to validate the imbalance of BDNF/TrkB signalling linked to SGN loss.

Author Response

The authors are grateful to Reviewer 2 for providing critical comments and critiques to help improve the quality of the manuscript.

  1. The term neurovascular should be defined appropriately: how the neuron can make a blood connection. Brief about the role of the glial cells in neurovascular dysfunction. It may provide a clear picture of how Inflammation (neuroinflammation) participates in amyloid cardiomyopathy, leading to myocardial infarction followed by heart failure during Age-related AD, as Inflammation (neuroinflammation) plays a role significant role in all types of cardiomyopathies, AD-allied pathologies, and BBB permeability.

This is a very important comment and we appologize for the lack in information. We included a comment : “The BBB is part of the so-called neurovascular unit. The neuron connects with the bloodstream through the neurovascular unit, which consists of neurons, endotheliocytes, and astrocytes [30].The latter cells are centrally positioned to mediate interactions between neurons and the cerebral vasculature to maintain the BBB integrity and neurovascular coupling. In AD, the inflammation within the neurovascular unit promotes the apoptosis of astrocytes, shutting down the neuronal food supply and causing BBB impairment and dysregulation. Moreover, the Aβ impact on the vascular component is defined by the Aβ deposition not only in the heart but in blood vessels, leading to a propensity for vessel rupture [31]. This condition was previously described as cerebral amyloid angiopathy a common cause of lobar intracerebral hemorrhage and cognitive impairment [32].“

  1. A few more lines should be added about the cardiac status of the AD patients/animal models and amyloid cardiomyopathy related to peripheral and autonomic neuropathy.This comment was especially appreciated, helping much to give the bigger picture of the disease. We added the following comment (L 214ff ): Shin, S. C. and J. Robinson-Papp (2012). "Amyloid neuropathies." The Mount Sinai journal of medicine, New York 79(6): 733-748.Gupta, N, Retnaswami, S, Malligurki, C, Rukmani, R, & ... (2017). Autonomic dysfunction in patients with Alzheimer's disease. Vol 
  2.  
  3. Serhiyenko, V. A. and A. A. Serhiyenko (2018). "Cardiac autonomic neuropathy: Risk factors, diagnosis and treatment." World Journal of Diabetes 9(1): 1-24.
  4. Liao, R. and J. E. Ward (2017). "Amyloid Cardiomyopathy: Disease on the Rise." Circulation research 120(12): 1865-1867.
  5. Embedding the relevant literature:
  6. It is well-known that AD patients often develop amyloid cardiomyopathy and peripheral and autonomic neuropathies (Shin and Robinson-Papp, 2012; Liao and Ward, 2017). The latter pathology may lead to a cardiac autonomic dysfunction/neuropathy, manifesting as sinus tachycardia, exercise intolerance, orthostatic hypotension, abnormal blood pressure regulation, dizziness, asymptomatic myocardial ischemia, and infarction (Gupta et al., 2017; Serhiyenko and Serhiyenko, 2018).   
  7.  
  8. Although the risk factors of hypertension, obesity etc.mentioned here are the prominent risk factors for cardiovascular diseases, mechanistic relations to the  BBB permeability and AD should be provided, like how the BBB permeability affects the Aβ accumulation/deposition, which could be based on the Aβ transportation/accumulation in the extracellular region of cardiomyocytes, leading to myocardial infarction/hypertrophy and its associated abnormalities.

We apologize for missing this part. We added the following explanatory sentences: “It is well known that AD, a neurodegenerative disorder, is characterized by chronic brain inflammation, which promotes the BBB dysfunction and increases its permeability [59]. Therefore, it might be hypothesized that this situation could trigger the Aβ permeation across the BBB to the bloodstream. Additionally, the reduced levels of low-density lipoprotein receptor at the BBB contributes to the elevated Aβ concentrations in plasma [60]. As a result, the subsequent Aβ accumulation and deposition occur in the extracellular region of cardiomyocytes, leading to heart pathologies, such as myocardial infarction, cardiomyopathy, and heart failure. Our hypothesis is also confirmed by the preliminary findings of Trancone et al, where the authors observed secondary amyloidosis of the heart in patients with AD [61].”

  1. One of the exciting para/lines (line 378-382) that links the ARHL and AD: whether Tauopathy is the root link between ARHL and AD?We agree that this hypothesis is worth to be addressed in more detail and modified the paragraph as follows: 
  2. “The authors want to stress in that context that consistent with the central dogma in AD investigation - tauopathy preceding Aβ- deposition[105]-, as striking evidence for early NFT formation, mainly consistent of hyperphosphorylated tau protein, has been steadily reported to occur in the central auditory pathway long before brain Aβ- deposition [106]. Instead of ARHL being one factor inducing AD tauopathy and APP deposition may be the root link between ARHL and AD, giving rise to both pathologies. The correlations between tauopathy – AD - ARHL need to be elucidated in future studies to identify the origin of the cascade and to establish appropriate therapy strategies.”
  3.  
  4. Line 401, the neurotrophins: indicateall neurotrophins or specific.- we included the respective neurotrophins the reference refers to (BDNF and NT-3) and added a reference on GDNF
  5.  
  6.  
  7. More evidences are required to validate the imbalance of BDNF/TrkB signalling linked to SGN loss.You are correct. There is no direct proof that ARHL related SGN decline is correlated to BDNF/trkB imbalance. All relevant literature we refer to give an indirect statement: that BDNF/trk-Agonists improve SGN survival in a ARHL scenario. We apologize for the simplified presentation of the facts and have rewritten the paragraph as follows: 
  8. Of note, AD shows imbalance of specific paralleling neurotrophic factors and receptors and the same is indirectly seen in ARHL where BDNF protein is drastically reduced in peripheral and central auditory projections [hier muss die 114 hin] and BDNF and trkB-agonists improve SGN survival [for review see leake at el, ], [114]implicating an initial imbalance.

Round 2

Reviewer 2 Report

the authors have addressed all my questions and I have no further.